⊙ | **Open Peer Review** | Clinical Microbiology | Research Article

# Development of an antigen detection test kit (Melioidosis-ATK) for point-of-care diagnosis of melioidosis

Akarin Intaramat,[1] Hasyanee Binmaeil,[2] Thanakon Bunsong,[2] Saowarat Deekae,[3] Sunee Chayangsu,[4] Mayuree Fuangthong,[1] Wang Nguitragool,[5] Mary N. Burtnick,[2,6] Paul J. Brett,[2,6] Narisara Chantratita[2,7]

**ABSTRACT** Melioidosis is a fatal infectious disease caused by the environmental bacterium *Burkholderia pseudomallei*. Although it is highly endemic in tropical regions, melioidosis remains underdiagnosed. The disease presents with diverse clinical manifestations, but diagnosis relies on time-consuming bacterial culture. Many patients with poor outcomes are referred from community hospitals, where culture facilities are unavailable. Early and accurate diagnosis is critical for timely treatment and improved outcomes. We developed an antigen detection test kit (Melioidosis-ATK) that targets the 6-deoxy-heptan capsular polysaccharide (CPS) of *B. pseudomallei*. The diagnostic performance of the assay was evaluated using 88 clinical samples, including blood culture broth (obtained from incubated blood culture bottles) ($N$ = 17), sputum ($N$ = 33), pus ($N$ = 17), urine ($N$ = 14), and other body fluids ($N$ = 7) from 88 patients with melioidosis, 204 samples from 195 patients with other bacterial infections, and 38 samples from 38 patients with negative blood culture. The assay demonstrated a limit of detection of $4.33 \times 10^3$ CFU/mL for *B. pseudomallei* and 0.2 ng/mL for purified CPS. The Melioidosis-ATK demonstrated high diagnostic performance, with a sensitivity of 100% for blood culture broth, 84.9% for sputum, 88.2% for pus, 100% for urine, and 85.7% for other body fluids. The assay also exhibited a specificity of 100% across all sample types evaluated. The Kappa coefficient of agreement between the antigen test and culture for all clinical specimens ranged from 0.77 to 1.00. These findings suggest that the Melioidosis-ATK is a promising point-of-care (POC) diagnostic tool for use in rural hospitals, offering rapid, simple, equipment-free testing with high performance and has the potential to improve patient outcomes.

**IMPORTANCE** Melioidosis is a life-threatening but underrecognized infectious disease caused by *Burkholderia pseudomallei*, which predominantly occurs in regions with limited diagnostic capacity. A definitive diagnosis typically relies on bacterial culture, which is slow, technically demanding, and often unavailable in resource-constrained settings. To address this problem, we developed a rapid antigen detection assay (Melioidosis-ATK) that targets the 6-deoxy-heptan capsular polysaccharide (CPS) of *B. pseudomallei* and evaluated its performance using multiple clinical specimen types. The test demonstrated high sensitivity and specificity and can be performed as a point-of-care (POC) test. By enabling prompt diagnosis and timely initiation of effective therapy, this assay represents a practical tool to enhance clinical management and reduce disease-associated mortality in endemic regions.

**KEYWORDS** *Burkholderia pseudomallei*, melioidosis, point-of-care test, antigen detection, diagnosis, Melioidosis-ATK, capsular polysaccharide

M elioidosis is a life-threatening infectious disease caused by *Burkholderia pseudo-mallei*, a motile, gram-negative, facultative intracellular, saprophytic bacterium

**Peer Reviewers** Tushar Shaw, M.S.Ramaiah University of Applied Sciences, Manipal, Karnataka, India; Saina Beitari, McGill University, Montreal, Quebec, Canada

Address correspondence to Narisara Chantratita, narisara@tropmedres.ac.

Akarin Intaramat and Hasyanee Binmaeil contributed equally to this article. Author order was determined based on decreasing seniority.

The authors declare no conflict of interest.

See the funding table on p. 13.

commonly found in soil and water in tropical and subtropical regions (1). Classified as a Tier 1 select agent by the U.S. Federal Select Agent Program, *B. pseudomallei* is highly endemic in northern Australia, South Asia, and Southeast Asia, particularly in countries such as Thailand, Laos, Cambodia, and Malaysia (2). The pathogen was identified in 48% of patients with septic arthritis in Thailand and in 74% of children with suppurative parotitis in Cambodia (3). High seroprevalence rates were also reported among tsunami survivors in Thailand (64%) and military personnel in Malaysia (54% and 66%) (3). In recent years, the geographical distribution of melioidosis has expanded beyond known endemic regions, with sporadic cases reported globally (4). Although melioidosis is not officially classified as a neglected tropical disease (NTD) by the World Health Organization (WHO) (2), its considerable disease burden and expanding global distribution underscore its emerging significance for public health. Human infections typically occur through percutaneous inoculation, inhalation, or ingestion of contaminated water or soil, and disease primarily occurs in individuals with underlying risk factors such as diabetes mellitus, chronic renal disease, or immunosuppression (2, 5).

The clinical manifestations of melioidosis are notoriously diverse, encompassing asymptomatic seroconversion, chronic localized infections, pulmonary disease, and fulminant septicemia with multiorgan failure. This wide clinical spectrum, often overlapping with presentations of tuberculosis, leptospirosis, or other respiratory infections, poses substantial diagnostic challenges, especially in endemic regions with limited laboratory infrastructure (2). Prompt initiation of effective antimicrobial therapy is critical in the management of melioidosis, which typically requires a prolonged treatment regimen comprising an intensive phase followed by eradication therapy. Melioidosis can lead to rapid clinical deterioration and death, with reported case fatality rates of up to 40% depending on such factors as clinical recognition, diagnostic capacity, antimicrobial therapy, and intensive care management (2, 6–8).

Bacterial culture from clinical specimens, including blood, urine, sputum, abscess fluid, and throat swabs, remains the gold-standard method for diagnosis. This process is commonly followed by standard biochemical testing, latex agglutination (9, 10), or matrix-assisted laser desorption/ionization time-of-flight mass spectrometry (MALDI-TOF MS) (11). While highly specific, culture-based methods have only 60% sensitivity and require biosafety level 3 (BSL-3) laboratory infrastructure, which is often unavailable in resource-limited, endemic areas (12). Furthermore, the method is time-consuming, often requiring several days, specialized expertise, and strict laboratory biosafety protocols. *B. pseudomallei* is frequently misidentified as a *Pseudomonas* species (1) and may produce false-negative results in cases of low bacterial load or prior antibiotic administration (13).

Several serological assays have been developed for the diagnosis of melioidosis, including complement fixation, indirect hemagglutination assay (IHA), indirect fluorescent antibody (IFA), enzyme-linked immunosorbent assays (ELISA), and an Hcp1-based immunochromatography test (Hcp1-ICT), which has demonstrated good performance; however, prospective studies across diverse geographic regions remain limited (14). Among these, the IHA has been widely used in countries where melioidosis is endemic due to its low cost. However, the performance of serodiagnostic tests is limited by high background seropositivity (15–17), low sensitivity in acute infections (18, 19), and cross-reactivity due to exposure to other environmental bacteria, making them unreliable as a sole diagnostic method (20). These limitations highlight the urgent need for diagnostic tools that are rapid, accurate, easy to implement, and suitable for use in both clinical laboratories and point-of-care (POC) settings.

Since there are currently no commercially available or licensed rapid antigen detection test kits for melioidosis, we developed the Melioidosis-ATK for POC detection of *B. pseudomallei* capsular polysaccharide (CPS) antigen in blood culture broth (obtained from incubated blood culture bottles) and directly from various clinical specimens. The test is designed as an immunochromatography strip, suitable for use in diagnostic laboratories, including those in resource-limited and endemic areas. We evaluated its diagnostic performance using 330 clinical specimens, including blood

culture broth, sputum, pus, urine, and other body fluids, from patients with culture-con-firmed melioidosis, other bacterial infections, and negative blood cultures (broth from incubated blood culture bottles with no microbial growth). The results were compared to those of bacterial culture followed by biochemical tests, which served as the gold standard.

## MATERIALS AND METHODS

### Preparation of heat-killed bacteria

Cultivation of *B. pseudomallei* was performed in a BSL3 laboratory. Heat-killed *B. pseudomallei* was prepared using a modified method as previously described (21). Briefly, *B. pseudomallei* strain K96243 was cultured in trypticase soy agar (TSA) at 37°C for 2 days, and 20 µL of the culture was subsequently added to 1 mL of sterile phosphate-buffered saline (PBS, pH 7.4). The bacterial suspension was centrifuged at 10,000 × $g$ for 10 min and washed twice with 1 mL of PBS. The pellet was resuspended in 1 mL PBS. To determine the bacterial count, 100 µL of the suspension was serially diluted 10-fold in PBS. From each dilution, 100 µL was plated in triplicate onto TSA and incubated at 37 °C for 16–18 h. The remaining bacterial suspension was heated at 80 °C for 1 h. Heat-killed bacteria (100 µL) were plated in duplicate on TSA for a sterility test.

### Antigen detection in blood culture broth

We tested the Melioidosis-ATK on 101 blood culture broth and 229 clinical specimens collected by hospital staff at Mukdahan Hospital, Roi Et Hospital, and Surin Hospital from 2021 to 2025. The blood culture broth was obtained from 17 patients with culture-con-firmed melioidosis, 46 patients with other bacterial infections, and 38 patients with clinical suspicion of melioidosis or other infections, but with negative blood cultures (broth from incubated blood culture bottles with no microbial growth).

Routine practice was to take 10 mL of blood for adults and inoculate this into a 30 mL blood culture bottle (Aerobic BacT/Alert FA Plus; Biomérieux, Marcy-l' Étoile, France). The bottle was incubated in an automated BacT/ALERT 3D or BacT/Alert VIRTUO instrument at 37°C. Once the instrument flagged the blood culture bottle as positive, it was obtained for Gram staining and sub-cultured for biochemical identification at the hospitals. The Melioidosis-ATK was performed on 50 µL of the fluid taken from blood culture bottles that were flagged as positive by the BacT/ALERT 3D or BacT/Alert VIRTUO instruments and identified as containing gram-negative bacteria by Gram stain. All procedures were conducted in a Class II biosafety cabinet.

### Antigen detection in clinical samples

A total of 229 anonymous clinical samples including sputum, pus, urine, and other body fluids (e.g., synovial fluid, peritoneal dialysis fluid, bile, knee fluid, fluid from abdominal, and pleural fluid) were directly used to evaluate the performance of Melioidosis-ATK. These specimens were categorized into the following groups: (i) 71 clinical samples collected from 71 patients with culture-confirmed melioidosis from any specimen, comprising sputum ($N = 33$), pus ($N = 17$), urine ($N = 14$), and body fluids ($N = 7$); (ii) 158 clinical samples obtained from 149 patients with culture-confirmed other infections, including sputum ($N = 64$), pus ($N = 5$), urine ($N = 80$), and body fluid ($N = 9$).

### Monoclonal antibodies and ELISA

The *B. pseudomallei* 6-deoxy-heptan (CPS)-specific mouse monoclonal antibodies (mAbs) 6G9, 4B11, and 2E2 (22–25) were used to develop the Melioidosis-ATK. The *B. pseudomal-lei* CPS-specific mouse mAb, LN2-3 (IgG1) developed in the Brett and Burtnick labs at the University of Nevada, Reno, was also used for control purposes. The activity of antibodies was assessed by an indirect ELISA using heat-killed *B. pseudomallei* K96243, as previously

described (26). In addition, purified CPS was prepared essentially, as previously described (27).

## Development of the Melioidosis-ATK

The antigen detection test was developed and optimized using four mAbs, namely, clones 2E2, 6G9, 4B11, and LN2-3, as capture antibodies and conjugate antibodies. The test was constructed with a nitrocellulose membrane (Cytiva, Massachusetts, USA). For preliminary evaluation, 2E2, 6G9, 4B11, LN2-3, or a mixture of 2E2 and 4B11 (750–1,000 µg/mL) was dotted onto the membrane. For constructing the test kits, antibodies were sprayed on a nitrocellulose membrane in a line pattern using the XYZ3060™ dispenser (Irvine, California, USA). Rabbit anti-immunoglobulin Y (IgY) (Sigma-Aldrich, St. Louis, USA) was used for the control line. The immobilized membrane was dried in a climate chamber (Memmert, Büchenbach, Germany) at 25°C and 35% relative humidity for 1 h, blocked with blocking buffer, and redried under identical conditions.

2E2 or 4B11 or a mixture of 2E2 and 4B11 mAbs was conjugated to a 40 nm colloidal gold particle (Arista, Allentown, PA) by passive absorption. Each mAb was added to the gold solution, and the resulting gold conjugates were resuspended in a solution of casein containing sucrose in sodium phosphate buffer. The chicken IgY-gold conjugate or mAb gold conjugates were opted and transferred to glass fiber filter GF33 (Whatman Schleicher & Schuell, Dassel, Germany). Subsequently, the materials were dried in a chamber at 25°C and 35% relative humidity for 24 h and stored in a dehumidifier cabinet until assembly into individual diagnostic test strips.

The final prototype Melioidosis-ATK consisted of a nitrocellulose membrane with immobilized test (T) and control (C) lines, incorporating an mAb specific to *B. pseudomallei* CPS, and a rabbit anti-IgY. The test strip also included a glass fiber impregnated with gold-labeled antibodies serving as signal reporters, an absorbent pad (Cytiva, Massachusetts, USA), and a sample pad (Cytiva, Massachusetts, USA) pretreated with Triton X-100 and polyvinylpyrrolidone (PVP) in Tris-HCl. The components were assembled with a 1 mm overlap, allowing the fluid to migrate through the strip via capillary action, onto a backing of plastic (Kenosha, Amstelveen, Netherlands). An absorbing pad was placed at the upper end of the nitrocellulose membrane. A sample pad was placed at the lower end of the strip. A glass fiber was inserted between the 2 sample pads. The incorporated materials were cut into 3-mm-wide individual test strips using a cutting module (Bio-Dot, California, USA).

## Melioidosis-ATK testing on blood culture broth

The Melioidosis-ATK assays were performed by mixing 50 µL of the blood culture broth with 50 µL of running buffer (PBS containing 0.5% Tween-20) in wells of a 96-well microtiter plate. Subsequently, the test strip was dipped into a mixture of sample and running buffer. The test and control lines were examined after 15 min. The assays were interpreted as positive if both T and C lines were visible, and they were interpreted as negative if only the C line was visible. The assays were considered invalid if the C line was not visible, and they were repeated. Any ambiguous results were viewed by 2 independent examiners reading the results in a blind manner. The performance of this assay was compared with the culture results.

## Melioidosis-ATK testing on urine and body fluids

The Melioidosis-ATK assays were performed by mixing 50 µL of urine or other body fluids with 50 µL of running buffer in the wells of a 96-well plate. Subsequently, the test strip was dipped into a mixture of sample and running buffer. The T and C lines were examined and interpreted as described above.

## Melioidosis-ATK testing on pus and sputum

Pus and sputum samples were diluted with an equal volume of PBS. Subsequently, 50 µL of the diluted sample was mixed with 50 µL of running buffer (PBS containing 0.5% Tween-20) in wells of a 96-well plate. A test strip was then dipped into a mixture of the sample and running buffer. The T and C lines were examined, and the assay results were interpreted as described above.

## Statistical analysis

Data were analyzed using IBM SPSS Statistics for Windows, version 29.0 (IBM Corp., Armonk, NY, USA), and GraphPad Prism, version 10.3.1 (GraphPad Software, San Diego, CA, USA). Kappa and McNemar tests were used to assess the agreement between different detection methods. Differences were considered significant when the *P*-value was <0.05. The accuracy, sensitivity, specificity, positive predictive value (PPV), and negative predictive value (NPV) of the Melioidosis-ATK were calculated separately for each sample type using culture as the reference method.

Sensitivity = $a/(a + c)$

Specificity = $d/(b + d)$

Positive predictive value (PPV) = $a/(a + b)$

Negative predictive value (NPV) = $d/(c + d)$

Accuracy = $(a + d)/(a + b + c + d)$

A kappa test was also used to determine concordance between Melioidosis-ATK results and culture results. The kappa value was determined using the following formula:

Kappa value (κ) = $(P_O - P_e) / (1 - P_e)$

$P_O = (a + d)/(a + b + c + d)$

$P_e = [(a + b)(a + c)+(b + d)(c + d)]/(a + b + c + d)^2$

where a = true positives, b = false positives, c = false negatives, and d = true negatives.

## RESULTS

### Evaluation of monoclonal antibodies for detecting heat-killed *B. pseudomallei* using Melioidosis-ATK strips

An indirect ELISA was conducted to evaluate the binding activity of mAbs 2E2, 4B11, and 6G9 to heat-killed *B. pseudomallei* K96243 across serial dilutions ranging from 1:10 to 1:102,400. As shown in Fig. 1A, the OD values at 450 nm demonstrated that mAbs 2E2 and 4B11 consistently exhibited higher signals than mAb 6G9, indicating their potential for applications that require robust antibody performance. To determine the optimal mAb for capture and conjugate components, the strip architecture was evaluated (Fig. 1B). Monoclonal antibodies 2E2, 4B11, 6G9, or their combinations were immobilized on the test region and conjugated to gold nanoparticles. The performance of the Melioidosis-ATK was assessed using heat-killed *B. pseudomallei* K96243 as a positive control and PBS as a negative control. As shown in Fig. 1C, mAb 4B11 produced the strongest red signal, visible to the naked eye, and was selected for use as the conjugate antibody in the Melioidosis-ATK.

Six different antibody combinations for the Melioidosis-ATK were tested, as shown in Fig. 1D, including (i) 4B11 mAb as both conjugate and capture antibodies; (ii) 4B11 as conjugate and 2E2 as capture; (iii) 4B11 as conjugate and a mixture of 4B11 and 2E2 as capture; (iv) a mixture of 4B11 and 2E2 as both conjugate and capture antibodies; (v) 2E2 as conjugate and a mixture of 4B11 and 2E2 as capture; and (vi) 4B11 as conjugate and LN2-3 as capture. The limit of detection (LOD) was evaluated with all antibody combinations (i–vi) using heat-killed *B. pseudomallei* K96243. The assay was performed using running buffer spiked with heat-killed *B. pseudomallei* at bacterial concentrations ranging from $4.33 \times 10^5$ to $4.33 \times 10^1$ CFU/mL, with running buffer alone serving as the negative control. The LOD was defined as the lowest bacterial concentration consistently producing a visible red test line. The LOD across all 6 antibody combinations was $4.33 \times 10^3$ CFU/mL (Fig. 1D).

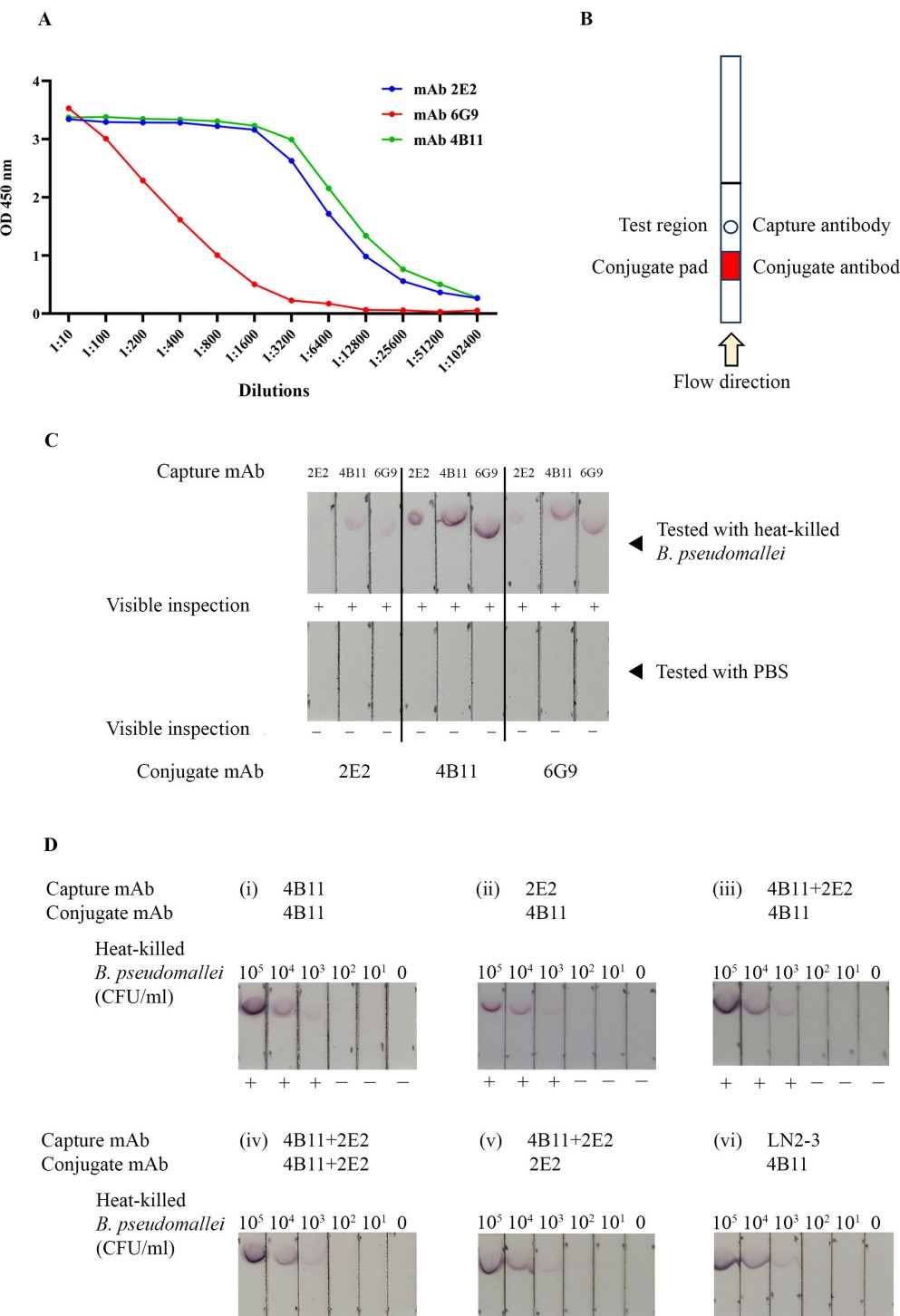

**FIG 1** Optimization of the Melioidosis-ATK. (A) Reactivity of CPS-specific monoclonal antibodies against heat-killed *B. pseudomallei,* as determined by indirect ELISA. (B) Schematic representation of the strip prototype with a dot pattern. (C) Optimization of capture and conjugate antibodies. (D) Limit of detection of the strip prototype in 6 different formats. The assay with different antibody combinations was tested using running buffer (PBS containing 0.5% Tween-20) spiked with heat-killed *B. pseudomallei* at bacterial concentrations ranging from $4.33 \times 10^5$ to $4.33 \times 10^1$ CFU/mL, with running buffer alone serving as the negative control. +, positive result; −, negative result.

## Preliminary evaluation of six Melioidosis-ATK prototypes using urine samples

A preliminary evaluation of the 6 Melioidosis-ATK prototypes was performed using 50 urine samples from 50 culture-confirmed melioidosis patients (from any specimen) and 25 patients with other bacterial infections. Using the patients as the denominator, the diagnostic sensitivity ranged from 34% to 44% across the 6 antibody combinations (Table 1). All test formats demonstrated 100% specificity compared to the reference culture method. The format (iii) using 4B11 mAb as the conjugate antibody and a mixture of 4B11 and 2E2 mAbs as the capture antibody demonstrated the highest diagnostic performance, with a sensitivity of 44% and 100% specificity.

We evaluated six Melioidosis-ATK prototypes using urine samples from patients from whom *B. pseudomallei* had been isolated from blood, sputum, pus, urine, or other body fluids. Among patients with bloodstream infections ($N$ = 34), sensitivities ranged from 29.4% to 38.2%, with format (iii) demonstrating the highest sensitivity at 38.2%. For patients with urine isolates ($N$ = 5), all six formats achieved 100% sensitivity (Table 2). Among patients with sputum isolates ($N$ = 2), all formats showed 50% sensitivity. In patients with pus isolates ($N$ = 9), sensitivities varied widely, with formats (iii), (iv), and (vi) showing the highest sensitivity at 33.3%, while format (i) showed no detection. All six test formats demonstrated 100% specificity across the corresponding negative samples tested (Table 2). Since format (iii) demonstrated the highest sensitivity for bloodstream and pus infections while maintaining 100% specificity across all specimen types, it was selected as the final Melioidosis-ATK prototype for further evaluation.

## Development of the Melioidosis-ATK

A mixture of the 2E2 and 4B11 mAbs was immobilized on the test (T) line as the capture antibodies, while rabbit anti-IgY was immobilized on the control (C) line. Gold-labeled 4B11 mAb and IgY were used as detection labels and were pre-applied to the glass fiber. The final Melioidosis-ATK prototype is shown in Fig. 2A. The test procedure is illustrated in Fig. 2B.

The LOD of the prototype Melioidosis-ATK was assessed using PBS spiked with heat-killed *B. pseudomallei* K96243 ranging from $4.33 \times 10^5$ to $4.33 \times 10^1$ CFU/mL, with running buffer alone serving as the negative control. The LOD was defined as the lowest bacterial concentration producing a positive result and was determined to be $4.33 \times 10^3$ CFU/mL (Fig. 2C). In addition, the assay demonstrated an LOD of 0.2 ng/mL for purified CPS in PBS.

The Melioidosis-ATK was evaluated using various types of clinical specimens including blood culture, sputum, pus, urine, and other body fluids. The tests showed positive and negative results across multiple specimen types (Fig. 2D), indicating its potential utility for detecting *B. pseudomallei* CPS directly in the different clinical samples.

**TABLE 1** Strip prototype formats and their performance for *B. pseudomallei* detection in urine samples from 50 patients with culture-confirmed melioidosis (from any specimen) and 25 patients with other bacterial infections[a,b]

| Strip prototype format | Capture mAb | Conjugate mAb | No. of positive urine samples (% sensitivity) (total = 50 patients) | No. of negative urine samples (% specificity) (total = 25 patients) |
|---|---|---|---|---|
| i | 4B11 | 4B11 | 17 (34%) | 25 (100%) |
| ii | 2E2 | 4B11 | 17 (34%) | 25 (100%) |
| iii | 4B11 + 2E2 | 4B11 | 22 (44%) | 25 (100%) |
| iv | 4B11 + 2E2 | 4B11 + 2E2 | 21 (42%) | 25 (100%) |
| v | 4B11 + 2E2 | 2E2 | 19 (38%) | 25 (100%) |
| vi | LN2-3 | 4B11 | 21 (42%) | 25 (100%) |

[a]Other bacterial infections included *Acinetobacter baumannii* ($N$ = 2), *Escherichia coli* ($N$ = 8), *Enterococcus faecalis* ($N$ = 1), *Morganella morganii* ($N$ = 1), *Klebsiella aerogenes* ($N$ = 1), *Klebsiella pneumoniae* ($N$ = 2), *Pseudomonas aeruginosa* ($N$ = 2), *Salmonella* spp. ($N$ = 2), and *Staphylococcus aureus* ($N$ = 6).
[b]Sensitivity and specificity were calculated using patients as the denominator.

**TABLE 2** Sensitivity and specificity of six Melioidosis-ATK strip prototypes tested on urine samples from 50 melioidosis patients with *B. pseudomallei* (confirmed by *B. pseudomallei* isolation from any specimen) and 25 patients with other bacterial infections[a]

| Specimens with *B. pseudomallei* isolation | Total no. of patients (N = 50) | No. of Melioidosis-ATK-positive urine samples (% sensitivity) and strip formats | | | | | | Total no. of patients (N = 25) | No. of Melioidosis-ATK-negative urine samples (% specificity) and strip formats | | | | | |
|---|---|---|---|---|---|---|---|---|---|---|---|---|---|---|
| | | i | ii | iii | iv | v | vi | | i | ii | iii | iv | v | vi |
| Blood | 34 | 11 (32.4) | 10 (29.4) | 13 (38.2) | 12 (35.3) | 12 (35.3) | 12 (35.3) | 15 | 15 (100) | 15 (100) | 15 (100) | 15 (100) | 15 (100) | 15 (100) |
| Sputum | 2 | 1 (50) | 1 (50) | 1 (50) | 1 (50) | 1 (50) | 1 (50) | 3 | 3 (100) | 3 (100) | 3 (100) | 3 (100) | 3 (100) | 3 (100) |
| Pus | 9 | 0 (0) | 1 (11.1) | 3 (33.3) | 3 (33.3) | 1 (11.1) | 3 (33.3) | 4 | 4 (100) | 4 (100) | 4 (100) | 4 (100) | 4 (100) | 4 (100) |
| Urine | 5 | 5 (100) | 5 (100) | 5 (100) | 5 (100) | 5 (100) | 5 (100) | 2 | 2 (100) | 2 (100) | 2 (100) | 2 (100) | 2 (100) | 2 (100) |
| Body fluid | 0 | ND[b] | ND | ND | ND | ND | ND | 1 | 1 (100) | 1 (100) | 1 (100) | 1 (100) | 1 (100) | 1 (100) |

[a]Sensitivity and specificity were calculated using patients as the denominator.
[b]ND, not done.

## Evaluation of the Melioidosis-ATK for diagnosing melioidosis using blood culture broth

The diagnostic performance of the Melioidosis-ATK was evaluated using a total of 101 blood culture samples, including 17 from patients with culture-confirmed melioidosis, 46 from patients with other infections, and 38 patients with clinical suspicion of melioidosis or other infections, but with negative blood cultures (broth from incubated blood culture bottles with no microbial growth). The Melioidosis-ATK demonstrated 100% sensitivity and specificity for detecting *B. pseudomallei* bloodstream infections with no false-positive or false-negative results (Table 3). Other microorganisms isolated from blood cultures obtained from 46 patients included *Acinetobacter baumannii* (N = 2), *Enterobacter cloacae* (N = 1), *Escherichia coli* (N = 4), *Klebsiella pneumoniae* (N = 3), *Roseomonas mucosa* (N = 1), *Salmonella* sp. (N = 1), *Bacillus* sp. (N = 3), *Corynebacterium afermentans* (N = 1), *Corynebacterium* spp. (N = 3), *Corynebacterium striatum* (N = 1), *Enterococcus casseliflavus* (N = 1), *Staphylococcus aureus* (N = 4), *Staphylococcus capitis* (N = 2), *Staphylococcus epidermidis* (N = 3), *Staphylococcus haemolyticus* (N = 5), *Staphylococcus hominis* (N = 3), *Staphylococcus sciuri* (N = 1), *Streptococcus agalactiae* (N = 3), *Streptococcus dysgalactiae* (N = 1), *Streptococcus mitis* (N = 1), *Streptococcus parasanguinis* (N = 1), and *Candida parapsilosis* (N = 1) (Table S1).

## Evaluation of the Melioidosis-ATK for diagnosis of melioidosis using various clinical samples

The diagnostic performance of the Melioidosis-ATK was evaluated using samples obtained from a total of 220 patients, including 71 patients with culture-confirmed melioidosis (from any specimen) and 149 patients with culture-confirmed other infections. Using the patient as the denominator, the Melioidosis-ATK demonstrated a sensitivity of 88.7% and a specificity of 100% among patients with other infections (Table 4).

The diagnostic performance of the Melioidosis-ATK was further evaluated using specimen type as the denominator, as shown in Table 5. A total of 229 clinical specimens were tested, including sputum (N = 33), pus (N = 17), urine (N = 14), and body fluids (N = 7) from 71 patients with culture-confirmed melioidosis, as well as sputum (N = 64), pus (N = 5), urine (N = 80), and body fluids (N = 9) from 149 patients with culture-confirmed other infections. The Melioidosis-ATK demonstrated high diagnostic sensitivity across specimen types, with sensitivities of 84.9% for sputum, 88.2% for pus, 100% for urine, and 85.7% for other body fluid samples. No false-positive results were observed across all

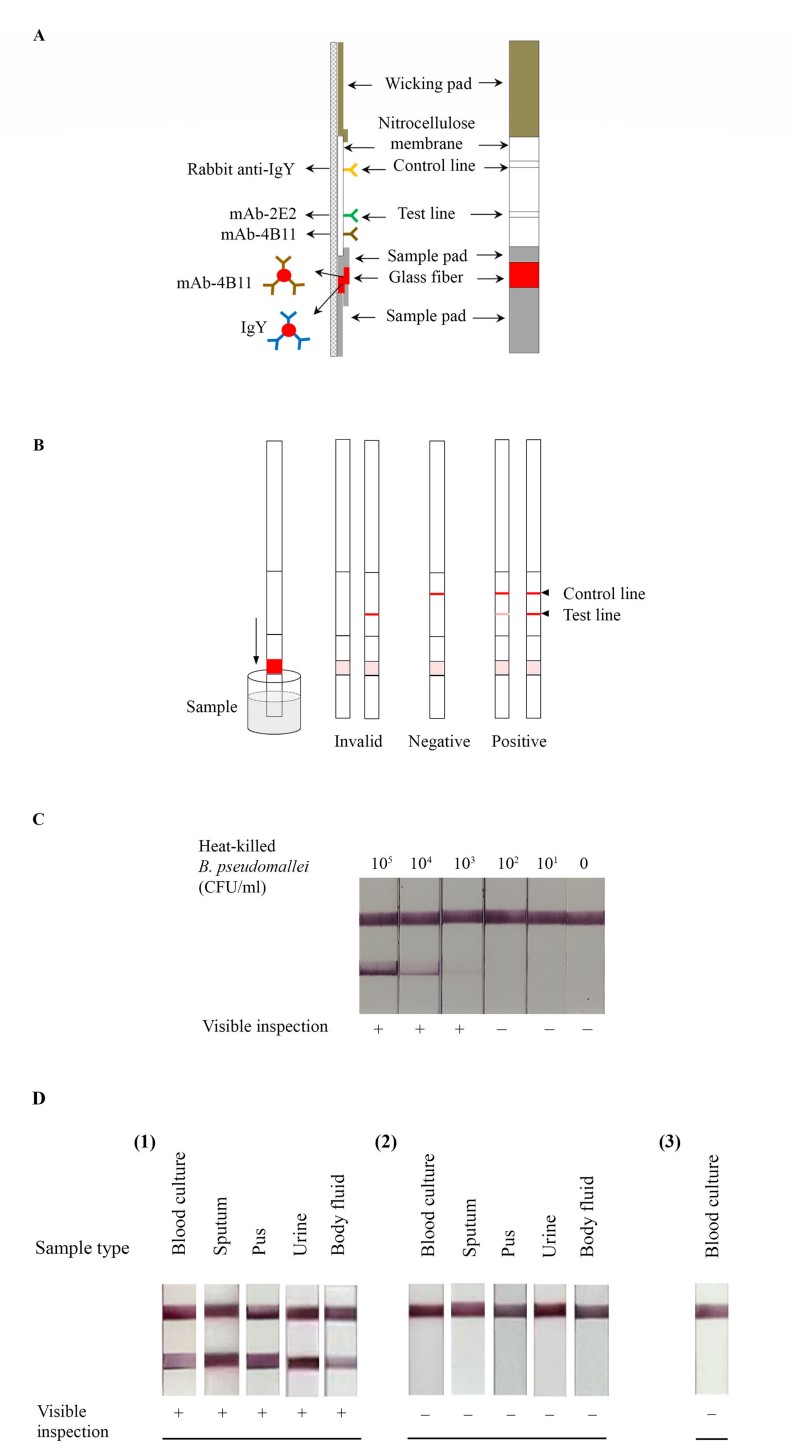

**FIG 2** Development of the Melioidosis-ATK. (A) Final prototype of the Melioidosis-ATK. The 4B11 mAb was used as the conjugate antibody, while mixture 2E2 and 4B11 mAbs served as capture antibodies. A chicken IgY gold conjugate was applied to react with rabbit anti-IgY at the control region to indicate the proper test function. (B) Schematic representation of the test procedure. (C) Determination of the LOD for the Melioidosis-ATK (line pattern). The assay was performed using running buffer (PBS containing 0.5% Tween-20) spiked with heat-killed *B. pseudomallei* at bacterial concentrations ranging from $4.33 \times 10^5$ to $4.33 \times 10^1$ CFU/mL, with running buffer alone serving as the negative control. (D) Representative results obtained using the Melioidosis-ATK with various clinical specimens. Negative blood culture refers to broth from incubated blood culture bottles with no microbial growth. +, positive result; −, negative result.

specimen types, while false-negative results were limited to sputum ($N = 5$), pus ($N = 2$), and body fluid ($N = 1$).

The Melioidosis-ATK showed 100% specificity across all specimen types when evaluated against samples from patients with other infections. None of the organisms isolated from these non-melioidosis specimens (Table S1) tested positive using the Melioidosis-ATK. Agreement with the culture method, as measured by Cohen's kappa coefficient, ranged from 0.77 to 1.00 (sputum = 0.88; pus = 0.77; urine = 1.00; and body fluid = 0.87).

## DISCUSSION

Currently, melioidosis is confirmed by culture, either through direct inoculation of clinical specimens (e.g., blood, urine, and pus) on selective media or by subculture and biochemical identification using broth from blood culture bottles (28–30). While culture remains the gold standard, it is slow and requires specialized facilities, delaying timely treatment. Early and accurate diagnosis is critical for improving outcomes, and rapid antigen detection assays provide clear advantages, including earlier detection, ease of use without specialized equipment, and suitability for resource-limited settings.

In this study, we developed a rapid POC Melioidosis-ATK targeting the 6-deoxy-heptan CPS antigen of *B. pseudomallei* (22–25). Optimization of membranes, buffer composition, antibody combinations, and antibody/gold conjugate concentrations resulted in clear test and control lines. With PBS containing 0.5% Tween-20 as the running buffer, the assay produced no false positives, and specific membranes provided consistent flow with sharp line development after 15 min, confirming reliable diagnostic performance. Among the 4 monoclonal antibodies tested, 4B11 was identified as the most effective conjugate antibody, and a combination of 4B11 and 2E2 was selected as optimal for the antigen capture pair. In preliminary testing with urine samples from culture-confirmed melioidosis cases, the kit achieved high sensitivity in all 6 test formats, particularly when *B. pseudomallei* was isolated from the same specimen type used for antigen testing. These findings provide proof of concept for clinical use.

Although initial testing of the Melioidosis-ATK directly with serum and plasma yielded low sensitivities, subsequent evaluation of the format (iii) prototype with urine samples from culture-confirmed patients and controls demonstrated high sensitivity and specificity compared with culture. We hypothesize that the Melioidosis-ATK could also be used with other specimens, including blood culture broth, sputum, pus, and other body fluids. The format (iii) prototype Melioidosis-ATK demonstrated an LOD of $4.33 \times 10^3$ CFU/mL using heat-killed *B. pseudomallei* and 0.2 ng/mL using CPS. The LOD of Melioidosis-ATK was lower than the previously reported detection of median bacterial loads in clinical specimens: $1.5 \times 10^4$ CFU/mL in urine, $1.1 \times 10^5$ CFU/mL in respiratory secretions, and $1.1 \times 10^7$ CFU/mL in pus (31). These findings indicate that the sensitivity of the assay is sufficient for direct detection of *B. pseudomallei* in different clinical specimens, with the exception of blood where the median level of bacteria was reported to be ~1.1 CFU/mL (31). Assay optimization was performed using heat-killed *B. pseudomallei* for biosafety and standardization purposes. Optimization using live organisms or freshly cultured colonies may influence assay performance as bacterial viability, growth

**TABLE 3** Diagnostic performance of Melioidosis-ATK using prospectively collected blood culture samples, compared with blood culture results[a]

| Type of sample | Melioidosis-ATK vs culture | | | | % Sensitivity (95% CI) | % Specificity using samples with other infections (95% CI) | % Specificity using samples with negative blood cultures (95% CI) | % PPV (95% CI) | % NPV (95% CI) | % Accuracy (95% CI) | Kappa (95% CI) |
|---|---|---|---|---|---|---|---|---|---|---|---|
| | TP | FN | FP | TN | | | | | | | |
| Blood culture fluid | 17 | 0 | 0 | 46 | 100 (17/17) (80.5–100) | 100 (46/46) (92.3–100) | 100 (38/38) (90.8–100) | 100 (17/17) (80.5–100) | 100 (46/46) (90.8–100) | 100 (63/63) (93.5–100) | 1.00 (1.00–1.00) |

[a]The analysis included 17 patients with melioidosis, 46 patients with other infections, and 38 patients with clinical suspicion of melioidosis or other infections, but with negative blood cultures (broth from incubated blood culture bottles with no microbial growth). TP, true positive; FN, false negative; FP, false positive; TN, true negative; PPV, positive predictive value; NPV, negative predictive value; Kappa, Cohen's Kappa (κ); CI, confidence interval.

**TABLE 4** Diagnostic performance of Melioidosis-ATK in 220 various clinical specimens compared with culture[a]

| Test | Total no. of samples | % Sensitivity (95% CI) | % Specificity using samples with other infections (95% CI) |
|---|---|---|---|
| Melioidosis-ATK | 220 | 88.7 (63/71) (79.0–95.0) | 100 (149/149) (97.6–100) |

[a]The results of the Melioidosis-ATK were evaluated in 71 patients with culture-confirmed melioidosis from any specimen and 149 patients with other infections. Sensitivity and specificity were calculated using patients as the denominator.

phase, and surface antigen expression, particularly the capsular polysaccharide, may differ from heat-killed cells and affect antigen accessibility. Nevertheless, the observed LOD using heat-killed organisms remains within clinically relevant bacterial ranges, supporting the feasibility of the assay for use across diverse clinical specimen types.

This study evaluated the Melioidosis-ATK using prospectively collected blood culture broth from patients with confirmed melioidosis (*N* = 17), patients with other infections (*N* = 46), and patients with clinical suspicion of melioidosis or other infections, but with negative blood cultures (broth from incubated blood culture bottles with no microbial growth) (*N* = 38) from Surin Hospital, Thailand. The assay demonstrated 100% sensitivity and specificity in perfect agreement with the results of blood culture broth followed by biochemical identification (κ = 1.0) (32). Similar performance has been reported with the InBios Active Melioidosis Detect (AMD) RDT using blood culture broth having a sensitivity of 96.5%–99% and specificity of 100% (33–35), although direct testing in whole blood yielded poor sensitivity (16.7%) (36). Because bacterial loads in blood are low (often 1.1 CFU/mL) (31), direct testing on blood samples is of limited use. However, the use of blood culture broth after short incubation enhances detection within 24–48 h, which is faster than conventional culture (37). These findings suggest that the Melioidosis-ATK is a highly accurate and practical diagnostic for blood culture broth and would be particularly useful in resource-limited settings such as in community hospitals, where it could enable earlier initiation of targeted therapy.

Across other specimen types, the Melioidosis-ATK demonstrated a strong diagnostic performance, with sensitivities of 84.9% for sputum, 88.2% for pus, 100% for urine, and 85.7% for other body fluid and 100% specificity for all non-*B. pseudomallei* infections. Urine testing showed excellent performance, exceeding the sensitivities reported by Woods et al. (86.7%) (35). Sensitivities for sputum, pus, and body fluid were comparable to those reported by Rizzi et al. (sputum 80%; pus 85.7%) (36) and higher than those reported by Woods et al. (sputum 33.3%, pus 47.1%, and body fluid 0%) (35). False negatives in sputum and pus may result from high sample viscosity. In this study, dilution with PBS improved consistency. However, it potentially could lower bacterial concentrations below the detection limit. Nevertheless, the Melioidosis-ATK developed

**TABLE 5** Sensitivity and specificity of Melioidosis-ATK using clinical samples[a]

| Type of sample (total samples) | Melioidosis-ATK vs culture | | | | % Sensitivity (95% CI) | % Specificity using samples with other infections (95% CI) | % PPV (95% CI) | % NPV (95% CI) | % Accuracy (95% CI) | Kappa (95% CI) |
|---|---|---|---|---|---|---|---|---|---|---|
| | TP | FN | FP | TN | | | | | | |
| Sputum (*N* = 97) | 28 | 5 | 0 | 64 | 84.9 (28/33) (68.1–94.9) | 100 (64/64) (94.4–100) | 100 (28/28) (87.7–100) | 92.8 (64/69) (85.1–96.6) | 94.9 (92/97) (88.4–98.3) | 0.88 (0.80–1.00) |
| Pus (*N* = 22) | 15 | 2 | 0 | 5 | 88.2 (15/17) (63.6–98.5) | 100 (5/5) (47.8–100) | 100 (15/15) (78.2–100) | 71.4 (5/7) (40.5–90.2) | 90.9 (20/22) (70.8–98.9) | 0.77 (0.48–1.00) |
| Urine (*N* = 94) | 14 | 0 | 0 | 80 | 100 (14/14) (76.8–100) | 100 (80/80) (95.5–100) | 100 (14/14) (76.8–100) | 100 (80/80) (95.5–100) | 100 (94/94) (96.2–100) | 1.00 (1.00–1.00) |
| Body fluid (*N* = 16) | 6 | 1 | 0 | 9 | 85.7 (6/7) (42.1–99.6) | 100 (9/9) (66.4–100) | 100 (6/6) (54.1–100) | 90 (9/10) (59.5–98.2) | 93.8 (15/16) (69.7–99.8) | 0.87 (0.63–1.00) |

[a]The results of the Melioidosis-ATK were compared with those of culture for *B. pseudomallei* (*N* = 71) and other infections (*N* = 158). Sensitivity and specificity were calculated using sample types as the denominator. TP, true positive; FN, false negative; FP, false positive; TN, true negative; PPV, positive predictive value; NPV, negative predictive value; Kappa, Cohen's Kappa (κ); CI, confidence interval.

herein maintained robust diagnostic accuracy for melioidosis detection across different specimen types.

The Melioidosis-ATK aligns with the WHO "ASSURED" criteria for rapid diagnostic tests: Affordable, Sensitive, Specific, User-friendly, Rapid, Equipment-free, and Deliverable (38) and demonstrates its potential impact in endemic settings. Further improvements include adapting the dipstick into a cassette format to enhance the biosafety and usability, as well as conducting operational evaluations to assess clinical adoption in multicenter settings. Combining antigen detection with antibody-based assays such as the Hcp1-ICT (14) may further enhance sensitivity and diagnostic coverage during acute and convalescent phases.

Several limitations should be noted. This study represents an initial clinical evaluation using residual clinical specimens rather than a large-scale prospective validation with predefined enrollment criteria. Although this approach reflects real-world clinical practice and enabled assessment across diverse specimen types, the limited number of melioidosis cases and certain specimen types may reduce statistical power and limit generalizability. Larger prospective studies with standardized enrollment and sampling strategies will therefore be required to further confirm diagnostic accuracy and clinical impact. Assay optimization and analytical evaluation were performed using heat-killed *B. pseudomallei* to ensure biosafety and reproducibility. We acknowledge that optimization using live organisms or cultured colonies, as well as variation in clinical specimen matrices, could influence antigen abundance, presentation, and assay performance. The limit of detection was determined under standardized conditions using PBS spiked with heat-killed bacteria or purified CPS and was not assessed separately across specimen types. Future studies employing live organisms and matrix-specific LOD evaluation are warranted, and head-to-head comparison with ELISA assays will be required to further strengthen assay validation. Finally, the limited performance observed with direct testing of serum or plasma indicates a need for further optimization or incorporation of culture enrichment strategies.

Despite these limitations, the Melioidosis-ATK demonstrated excellent sensitivity and specificity with blood culture broth and urine and good performance with sputum, pus, and other body fluids. It is rapid, standardized, and does not require specialized equipment or expertise, making it suitable for POC use in endemic regions. By enabling earlier diagnosis, particularly through urine testing, this assay could significantly improve the clinical management of melioidosis. While not a replacement for culture in drug susceptibility testing, the Melioidosis-ATK represents a promising tool for rapid case detection, surveillance, and timely treatment initiation.

## ACKNOWLEDGMENTS

We are grateful for the support of the staff at Mukdahan Hospital, Roi Et Hospital, and Surin Hospital and Department of Microbiology and Immunology, Faculty of Tropical Medicine, Mahidol University, Thailand. We thank Suporn Paksanont, Adul Dulsuk, and Rungnapa Phunpang for laboratory assistance and data collection.

This research project was funded by Mahidol University (Fundamental Fund: fiscal year 2024 by the National Science Research and Innovation Fund [NSRF]). The research project was partially supported by a postdoctoral fellowship award from Mahidol University, Thailand. This research was funded in part by the Wellcome Trust [220211/Z/20/Z].

For the purpose of Open Access, the author has applied a CC BY public copyright license to any Author Accepted Manuscript version arising from this submission.

A.I., H.B., and N.C. conceived and designed the experiments. T.B., S.D., and S.C. enrolled the patients and collected the clinical samples. A.I., H.B., and T.B. performed the experiments. H.B. and N.C. analyzed the data. A.I., H.B., M.F., M.N.B., P.J.B., and N.C. contributed reagents, materials, and analysis tools. A.I., H.B., M.F., W.N., and N.C. were responsible for project administration and supervision. H.B. and N.C. wrote the manu-

script. A.I., M.F., M.N.B., P.J.B., and N.C. reviewed and edited the manuscript. All authors have read and approved the manuscript.

## AUTHOR AFFILIATIONS

[1]Translational Research Unit, Chulabhorn Research Institute, Bangkok, Thailand

[2]Department of Microbiology and Immunology, Faculty of Tropical Medicine, Mahidol University Bangkok, Bangkok, Thailand

[3]Department of Medical Technology and Clinical Pathology, Surin Hospital, Surin, Thailand

[4]Department of Internal Medicine, Surin Hospital, Surin, Thailand

[5]Department of Molecular Tropical Medicine and Genetics, Faculty of Tropical Medicine, Mahidol University, Bangkok, Thailand

[6]Department of Microbiology and Immunology, University of Nevada, Reno School of Medicine, Reno, Nevada, USA

[7]Mahidol-Oxford Tropical Medicine Research Unit, Faculty of Tropical Medicine, Mahidol University, Bangkok, Thailand

## AUTHOR ORCIDs

Akarin Intaramat  http://orcid.org/0009-0003-1281-1038
Hasyanee Binmaeil  http://orcid.org/0000-0002-2129-5800
Mayuree Fuangthong  http://orcid.org/0000-0001-5684-9001
Wang Nguitragool  https://orcid.org/0000-0002-5484-7130
Mary N. Burtnick  http://orcid.org/0000-0001-6763-4348
Paul J. Brett  http://orcid.org/0000-0002-5555-6714
Narisara Chantratita  http://orcid.org/0000-0003-3906-7379

## FUNDING

| Funder | Grant(s) | Author(s) |
| --- | --- | --- |
| Mahidol University | Fundamental Fund 2024 | Narisara Chantratita |
| Mahidol University | Fundamental Fund 2024 | Wang Nguitragool |
| Mahidol University | Postdoctoral Fellowship Award | Hasyanee Binmaeil |
| Wellcome Trust | 220211/Z/20/Z | Narisara Chantratita |

## AUTHOR CONTRIBUTIONS

Akarin Intaramat, Conceptualization, Data curation, Formal analysis, Investigation, Methodology, Resources, Validation, Writing – original draft, Writing – review and editing | Hasyanee Binmaeil, Data curation, Formal analysis, Investigation, Methodology, Project administration, Supervision, Validation, Visualization, Writing – original draft, Writing – review and editing | Thanakon Bunsong, Data curation, Formal analysis, Investigation, Methodology, Writing – review and editing | Saowarat Deekae, Investigation, Methodology, Resources, Writing – review and editing | Sunee Chayangsu, Data curation, Investigation, Methodology, Resources, Writing – review and editing | Mayuree Fuangthong, Investigation, Resources, Supervision, Writing – review and editing | Wang Nguitragool, Project administration, Writing – review and editing | Mary N. Burtnick, Data curation, Investigation, Methodology, Resources, Supervision, Writing – review and editing | Paul J. Brett, Data curation, Investigation, Methodology, Resources, Supervision, Writing – review and editing | Narisara Chantratita, Conceptualization, Data curation, Formal analysis, Funding acquisition, Investigation, Methodology, Project administration, Resources, Supervision, Validation, Visualization, Writing – original draft, Writing – review and editing

## DATA AVAILABILITY

The data supporting the findings of this study are available from the corresponding author upon reasonable request.

## ETHICS APPROVAL

This study was approved by the Ethics Committee of Faculty of Tropical Medicine, Mahidol University (MUTM 2021-050-01 and MUTM 2025-013-01), and Surin Hospital (43/2568).

## ADDITIONAL FILES

The following material is available online.

### Supplemental Material

**Table S1 (Spectrum02881-25-s0001.pdf).** Non-*B. pseudomallei* pathogens identified by culture from various clinical specimens (*N* = 204) and their results when tested with the Melioidosis-ATK.

### Open Peer Review

**PEER REVIEW HISTORY (review-history.pdf).** An accounting of the reviewer comments and feedback.

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
