## [Reviewer comments · Microbiology Spectrum]

Microbiology Spectrum

Development of an Antigen Detection Test Kit (Melioidosis-ATK) for Point-of-Care Diagnosis of Melioidosis

Akarin Intaramat, Hasayanee Binmaeil, Thanakon Bunsong, Saowarat Deekae, Sunee Chayangsu, Mayuree Fuangthong, Wang Nguitragee, Mary Burtnick, Paul Brett, and Narisara Chantratita

Corresponding Author(s): Narisara Chantratita, Mahidol University

Review Timeline:

Submission Date:	September 16, 2025
Editorial Decision:	January 29, 2026
Revision Received:	February 12, 2026
Accepted:	February 18, 2026

Editor: Eleanor Powell

Reviewer(s): Disclosure of reviewer identity is with reference to reviewer comments included in decision letter(s). The following individuals involved in review of your submission have agreed to reveal their identity: Tushar Shaw (Reviewer #1); Saina Beitari (Reviewer #2)

Transaction Report:

DOI: <https://doi.org/10.1128/spectrum.02881-25>

Re: Spectrum02881-25 (**Development of an Antigen Detection Test Kit (Melioidosis-ATK) for Point-of-Care Diagnosis of Melioidosis**)

Dear Prof. Narisara Chantratita:

Thank you for the privilege of reviewing your work. After receiving feedback from two reviewers, modifications are required before potential publication. Below you will find instructions from the Spectrum editorial office, and the reviewer comments.

Revision Guidelines

Sincerely,
Eleanor Powell
Editor
Microbiology Spectrum

Reviewer #1 (Comments for the Author):

1. The study has taken random samples for evaluation of the kit, A detailed evaluation and validation is required in near future
2. Also a overall sensitivity and specificity including positive cases detected by gold standard versus detected by Meliod TAK test should also be depicted in the results which is not clear.

Reviewer #2 (Comments for the Author):

Intaramat et al., in their paper entitled "Development of an Antigen Detection Test Kit (Melioidosis-ATK) for Point-of-Care Diagnosis of Melioidosis," describe in detail the development of a rapid antigen detection test for the diagnosis of melioidosis. This approach is highly practical for resource-limited settings where access to laboratory culture facilities is limited. The authors clearly outline the test development process and the different prototypes evaluated to achieve optimal diagnostic performance. Additionally, the inclusion of diverse specimen types-including blood, urine, sputum, and others-further strengthens the applicability of the assay. However, several points require clarification:

The optimization and validation of the Melioidosis-ATK were performed using heat-killed *Burkholderia pseudomallei*. It would be helpful for the authors to discuss whether optimization using intact live organisms or cultured colonies might have influenced assay performance, and whether the results would be expected to differ under those conditions.

In addition, it would be helpful to compare the diagnostic performance of the developed POC test, using the same sample set, against a commercial ELISA and to include a direct performance comparison.

As already discussed in the limitation, determining the LOD based on different sample matrices would further strengthen the study. Demonstrating that the LOD does not differ across specimen types would improve confidence in the assay's robustness and clinical applicability.

RESPONSE TO REVIEWERS

Reviewer #1

Comment 1:

The study has taken random samples for evaluation of the kit. A detailed evaluation and validation is required in near future.

Response:

We agree with the reviewer that a larger and more structured evaluation is essential to fully establish diagnostic accuracy. The present study represents an initial clinical evaluation using available clinical specimens rather than a large-scale prospective validation. This approach was intended to reflect real-world clinical practice and to provide preliminary evidence of assay feasibility across diverse specimen types. We have revised the Discussion to explicitly acknowledge this limitation and to clarify that larger prospective validation studies with predefined enrollment criteria are planned as future work (line 463 - 465).

Comment 2:

An overall sensitivity and specificity including positive cases detected by gold standard versus detected by Meliod-ATK test should also be depicted in the results which is not clear.

Response:

We thank the reviewer for this important suggestion. We have revised the Results section to clearly present overall diagnostic sensitivity and specificity of the Melioidosis-ATK in comparison with the gold standard (bacterial culture) methods. A summary table has been added to explicitly show concordance between bacterial culture–positive and –negative cases and Melioidosis-ATK results, improving clarity of diagnostic performance assessment. These

revisions are presented in Tables 3 and 5, with corresponding additions in lines 348 and 374–376.

Reviewer #2

Comment 1:

The optimization and validation of the Melioidosis-ATK were performed using heat-killed *Burkholderia pseudomallei*. It would be helpful for the authors to discuss whether optimization using intact live organisms or cultured colonies might have influenced assay performance, and whether the results would be expected to differ under those conditions.

Response:

We appreciate this insightful comment. We have expanded the Discussion to address the potential impact of using heat-killed bacteria during assay optimization in lines 416-422. While the use of heat-killed organisms ensured biosafety and experimental reproducibility, we explicitly acknowledge that antigen abundance or presentation may differ in live bacteria or cultured colonies. We have clarified this as a limitation and stated that future studies using live organisms under appropriate biosafety conditions will be required to further strengthen assay validation (line 466-471).

Comment 2:

It would be helpful to compare the diagnostic performance of the developed POC test, using the same sample set, against a commercial ELISA and to include a direct performance comparison.

Response:

The comparison with a commercial ELISA using the same sample set would be valuable. However, in routine hospital practice, ELISA is not used for the diagnosis of melioidosis, and

there are currently no commercially available CPS-specific ELISA kits for *B. pseudomallei*, thereby precluding such a comparison. This limitation has now been addressed in the Discussion (lines 472).

Comment 3:

As already discussed in the limitation, determining the LOD based on different sample matrices would further strengthen the study. Demonstrating that the LOD does not differ across specimen types would improve confidence in the assay's robustness and clinical applicability.

Response:

We agree with the reviewer that matrix-specific determination of the limit of detection would further strengthen confidence in assay robustness and clinical applicability. In the present study, the LOD was not assessed separately across different specimen matrices. This limitation has been clarified in the Discussion, and matrix-specific LOD evaluation has been highlighted as an important focus of future work (line 469-471).

Re: Spectrum02881-25R1 (**Development of an Antigen Detection Test Kit (Melioidosis-ATK) for Point-of-Care Diagnosis of Melioidosis**)

Dear Prof. Narisara Chantratita:

I'm pleased to share that your manuscript has been accepted, and I am forwarding it to the ASM production staff for publication. Your paper will first be checked to make sure all elements meet the technical requirements. ASM staff will contact you if anything needs to be revised before copyediting and production can begin. Otherwise, you will be notified when your proofs are ready to be viewed.

Sincerely,
Eleanor Powell
Editor
Microbiology Spectrum